# Alcohol-Induced Lysosomal Damage and Suppression of Lysosome Biogenesis Contribute to Hepatotoxicity in HIV-Exposed Liver Cells

**DOI:** 10.3390/biom11101497

**Published:** 2021-10-11

**Authors:** Moses New-Aaron, Paul G. Thomes, Murali Ganesan, Raghubendra Singh Dagur, Terrence M. Donohue, Kharbanda K. Kusum, Larisa Y. Poluektova, Natalia A. Osna

**Affiliations:** 1Department of Environmental Health, Occupational Health, and Toxicology, College of Public Health, University of Nebraska Medical Center, Omaha, NE 68198, USA; 2Research Service, Veterans Affairs Nebraska-Western Iowa Health Care System, Omaha, NE 68105, USA; paul.thomes@unmc.edu (P.G.T.); murali.ganesan@unmc.edu (M.G.); raghu.dagur82@gmail.com (R.S.D.); tdonohue@unmc.edu (T.M.D.J.); kkharbanda@unmc.edu (K.K.K.); 3Department of Internal Medicine, University of Nebraska Medical Center, Omaha, NE 68105, USA; 4Department of Pharmacology and Experimental Neuroscience, University of Nebraska Medical Center, Omaha, NE 68105, USA; lpoluekt@unmc.edu

**Keywords:** lysosome damage, lysosome biogenesis, HIV, ethanol metabolites, hepatotoxicity

## Abstract

Although the causes of hepatotoxicity among alcohol-abusing HIV patients are multifactorial, alcohol remains the least explored “second hit” for HIV-related hepatotoxicity. Here, we investigated whether metabolically derived acetaldehyde impairs lysosomes to enhance HIV-induced hepatotoxicity. We exposed Cytochrome P450 2E1 (CYP2E1)-expressing Huh 7.5 (also known as RLW) cells to an acetaldehyde-generating system (AGS) for 24 h. We then infected (or not) the cells with HIV-1_ADA_ then exposed them again to AGS for another 48 h. Lysosome damage was assessed by galectin 3/LAMP1 co-localization and cathepsin leakage. Expression of lysosome biogenesis–transcription factor, TFEB, was measured by its protein levels and by in situ immunofluorescence. Exposure of cells to both AGS + HIV caused the greatest amount of lysosome leakage and its impaired lysosomal biogenesis, leading to intrinsic apoptosis. Furthermore, the movement of TFEB from cytosol to the nucleus via microtubules was impaired by AGS exposure. The latter impairment appeared to occur by acetylation of α-tubulin. Moreover, ZKSCAN3, a repressor of lysosome gene activation by TFEB, was amplified by AGS. Both these changes contributed to AGS-elicited disruption of lysosome biogenesis. Our findings indicate that metabolically generated acetaldehyde damages lysosomes and likely prevents their repair and restoration, thereby exacerbating HIV-induced hepatotoxicity.

## 1. Introduction

Since the emergence of antiretroviral therapy, a significant decline in AIDS-related mortality among people living with HIV (PLWH) precipitated an upsurge in non-AIDS-related mortality (NARM). Comorbidities implicated in NARM among PLWH are cardiometabolic disorders, renal disease, mental illness, and liver disease [1]. While cardiometabolic disease ranks as the leading comorbidity among PLWH [2], liver disease remains the leading injury for NARM, accounting for approximately one-fifth of all cases [3,4,5]. This makes the management of liver disease a priority for HIV care delivery. Pathogenesis of HIV-related liver comorbidities is exacerbated by “second hits”, such as alcohol or co-infection with hepatotropic viruses [6]. However, the mechanism(s) by which alcohol induces liver injury during HIV infection is/are not entirely clear, which prevents improving a holistic treatment strategy. Thus, an understanding of the mechanisms of HIV- and alcohol-induced hepatotoxicity is the goal of this study.

We recently showed that in HIV-infected liver cells, ethanol and/or its metabolite acetaldehyde cause accumulation of HIV proteins to induce oxidant stress and subsequent apoptotic cell death [7]. This may lead to detrimental consequences and promote the development of end-stage liver diseases since HIV-particle-bearing apoptotic hepatocytes are engulfed by hepatic stellate cells (HSC) and macrophages (Mph), to induce profibrotic and proinflammatory events, leading to the progression of liver fibrosis [7].

In untreated liver cells, lysosomes degrade HIV proteins, thereby preventing their intracellular accumulation and subsequent oxidant stress [7,8]. While lysosome dysfunction by ethanol has been reported [9,10], it has not been implicated in the pathogenesis of HIV/alcohol-induced liver injury. Here, we characterized the contribution of acetaldehyde, the primary oxidation product of ethanol, and its ability to impair lysosome function in HIV-exposed liver cells. First, we investigated whether the alcohol-induced decline in cathepsin activities was accompanied by decreased lysosome numbers. Second, we determined whether acetaldehyde-induced lysosome leakage triggered the cathepsin decline. Lysosome damage/rupture causes the release of cathepsins and other hydrolases into the cytosol. While the etiology of lysosome damage and subsequent leakage are diverse, ROS production induces such damage [11]. Here, we used an in vitro acetaldehyde-generating system (AGS) that continuously generates physiologically relevant amounts of acetaldehyde and mimics ethanol metabolism in HIV-infected Huh 7.5 (also known as RLW) cells that overexpress cytochrome P450 2E1 (CYP2E1), but not alcohol dehydrogenase (ADH). AGS generates acetaldehyde by using ethanol as the primary substrate of yeast alcohol dehydrogenase in the presence of nicotinamide adenine dinucleotide (NAD^+^) as a co-factor. Exposure of cells to the AGS elevates ROS levels in RLW cells since ethanol is part of AGS, and RLW cells generate ROS by CYP2E1 upon ethanol exposure [7]. Hence, we postulated that in HIV-exposed RLW cells, the decrease in cathepsin activities is due to lysosome leakage triggered by enzymatically generated acetaldehyde and oxidant stress, which mimics ethanol metabolism in primary hepatocytes. Such leakage is detrimental if lysosome repair/replacement is inhibited as well. The activity and intracellular localization of Transcription Factor EB (TFEB), which activates genes that encode lysosomal proteins and lysosome biogenesis [12], is disrupted by chronic ethanol administration [13,14]. Here, we sought to determine whether HIV and AGS prevent TFEB translocation from cytosol to the nucleus of HIV-infected RLW cells. TFEB nuclear translocation activates genes involved in lysosome biogenesis. TFEB nuclear translocation/gene activation is prevented by its phosphorylation [15], by reduced proteasomal degradation in the cytosol [16], and by impaired TFEB trafficking; an increased expression of ZKSCAN3, a transcriptional repressor of autophagy [17], also affects lysosomal gene activation. We hypothesized that exposure of hepatocytes to HIV and ethanol and its metabolite, acetaldehyde, induces oxidant stress to trigger lysosome leakage and lysosome biogenesis suppression. The latter suppression blocks intracellular replacement of damaged lysosomes causing cytotoxicity during combined HIV and alcohol/acetaldehyde exposure to liver cells.

## 2. Materials and Methods

### 2.1. Reagents and Antibodies

High-glucose Dulbecco’s modified eagle medium (DMEM) and fetal bovine serum were purchased from Invitrogen (Waltham, MA, USA); Trizol was from Life Technologies (Carlsbad, CA, USA); primer probes, high-capacity reverse transcription kit, and real-time polymerase chain reaction (RT-PCR) reagents were from Applied Biosystems by Thermo Fisher Scientific, Carlsbad, CA, USA. Pan-caspase inhibitor (PCI) was obtained from Ubiquitin-Proteasome Biotechnologies (UBPBio) Inc. (Cat#F7110, Aurora, CO, USA), Nocodazole was from Sigma Aldrich (St. Louis, MO, USA, M1404). Primary antibodies were: (a) mouse monoclonal: anti-TFEB (Santa Cruz Biotechnology, Dallas, TX, USA, sc-166736), anti-cathepsin B (H-5) (sc-365558), anti-TATA binding protein (Millipore Sigma, Burlington, MA, USA, MAB3658), anti-beta-actin (Santa Cruz Biotechnology, Dallas, TX, USA, sc-69879), anti-LAMP1 (Santa Cruz Biotechnology, Dallas, TX, USA, sc-20011), anti-Acetylated α-tubulin (Millipore Sigma, Burlington, MA, USA, T6793), anti-α-tubulin (Millipore Sigma, Burlington, MA, USA, T9026); (b) rabbit monoclonal and polyclonal anti-TFEB, 4240S; anti-Phospho-TFEB (ser211), 37681S; anti-Tom20 (D8T4N), 42406; anti-Galectin-3/LGALS3 (D4I2R), #87985; anti-ZKSCAN3 (LSBio, Seattle, WA, USA, LS-C501696); (c) secondary antibodies: (I) IRDye 680RD Goat anti-Rabbit, C50317-02; IRDye 680RD Donkey anti-Mouse, C50520-02; (II) Goat anti-Mouse IgG (H + L) Cross-Adsorbed Secondary Antibody, Alexa Fluor 555, Carlsbad, CA, USA, A21422; and Donkey anti-Rabbit IgG (H + L) Highly Cross-Adsorbed Secondary Antibody, Alexa Fluor 488.

### 2.2. In Vitro Studies

Due to the paucity of primary human hepatocytes, we performed all experiments on hepatocyte-like RLW cells. These cells have reduced innate immunity and do not sufficiently oxidize (metabolize) ethanol because they lack ADH. To circumvent this limitation, we exposed RLW cells to an exogenous constant source of acetaldehyde (acetaldehyde-generating system, AGS), which consists of 0.02 EU yeast alcohol dehydrogenase (ADH), 2 mM nicotinamide adenine dinucleotide (NAD), and 50 mM ethanol. AGS is known to continuously generate a physiologically relevant amount of acetaldehyde without unwarranted toxicities [7]. HIV-1_ADA_ propagated on primary human macrophages was purified at University of Nebraska Medical Center (UNMC) [18]. RLW cells were pre-treated for 24 h with AGS, infected with HIV-1_ADA_, and then exposed again to AGS for another 48 h. HIV was removed after overnight treatment, and the cells were washed thoroughly with 1× phosphate-buffered saline (PBS) to remove extracellular HIV.

### 2.3. RNA Isolation and Real-Time PCR (RT-PCR)

RNAs encoding p53, cathepsin D, and LAMP1 were quantified by RT-PCR as previously described [7]. Briefly, total cellular RNA was isolated from cells using Trizol reagent. A two-step procedure was used, in which 200 ng RNA was reverse-transcribed to cDNA using the high-capacity reverse transcription kit. In the second step, the cDNA was amplified using TaqMan Universal Master Mix II with fluorescent-labeled primers (TaqMan gene expression systems). These were incubated in a Model 7500 qRT-PCR thermal cycler. The relative quantity of each RNA transcript was calculated by its threshold cycle (Ct) after subtracting that of the reference (GAPDH).

### 2.4. Proteasome Activities

Chymotrypsin (Cht-L) and trypsin-like (T-L) peptidase activities were detected by in vitro fluorometric assay as previously reported by our laboratory [19]. Fluorogenic peptide substrates N-succinyl-leu-leu-val-tyr-7-amido-4-methlycoumarin (suc-LLVY-AMC; UBPBio, Inc., Aurora, CO, USA) and boc-leu-ser-thr-arg-7-amido-4-methlycoumarin (boc-LSTR-AMC; UBPBio, Inc., Aurora, CO, USA) were used to measure chymotrypsin- and trypsin-like activities of the proteasome.

### 2.5. Cathepsin B and L Activities

Cathepsin B and L activities were measured as previously described [20], using Z-arg-arg-7-amido-4-methylcoumarin hydrochloride (cathepsin B) and L-phe-arg-7-amido-4-methylcoumarin hydrochloride (cathepsin L).

### 2.6. Whole-Cell Lysates and Nuclear/Cytosolic Fractionations

Whole-cell lysates were prepared in phosphorylation capture buffer (0.5 M EDTA, 2 M Tris, 20 mM Na_3_VO_4_, 200 mM Na_4_P_2_O_7_, 100 mM PMSF, 1 M NaF, 20% Triton X-100, and aprotinin, pH 7). The extraction of the nuclear and cytosolic components from RLW cells was performed using the NE-PER Nuclear and Cytoplasmic Extraction Kit (ThermoFisher Scientific, Carlsbad, CA, USA) based on the manufacturer’s protocol.

### 2.7. Immunoblotting

Immunoblotting was performed as described [21,22]; blots were developed using Odyssey infrared imaging system, which was also used to quantify the protein bands [23]. Equal (20 µg) amounts of protein were loaded in each lane. Beta-Actin or TATA-binding proteins (TBP) were used as loading controls to normalize the protein band densities.

### 2.8. Immunofluorescence

RLW Cells (130,000/well) were seeded onto coverslips, which were inserted into each well of a 12-well plate. Cells were infected with the HIV-1 virus (Multiplicity of infection, MOI = 0.1). After overnight incubation of cells with HIV, the virus-containing medium was removed and replenished with fresh media. Cells were incubated for another 48 h of AGS treatment; cells were washed with 1× PBS, fixed with 4% paraformaldehyde for 12 min at 37 °C, permeabilized with 0.1% Triton X-100 for 3 min at room temperature, and blocked for 30 min with 1% (g/vol) bovine serum albumin (BSA) in PBS. Cells were stained to study the colocalization of LAMP1:Galectin 3, Tom20:Cathepsin-B, and quantification of LAMP1. First, cells were incubated with primary antibodies for 1 h. The cells were then washed and incubated with the mixture of Alexa-Fluor-labeled secondary antibodies for 30 min. Nuclei were stained with DAPI. The coverslips were transferred to microscope slides for imaging by using LSM 710 confocal microscope. Immunostainings and colocalization were quantified for intensity using National Institute of Health (NIH) Image J program.

### 2.9. Statistical Analyses

Data were analyzed using GraphPad Prism v7.03 software (GraphPad, La Jolla, CA, USA). Data from at least three duplicate independent experiments were expressed as mean ± SEM. Comparisons among multiple groups were performed by one-way ANOVA, using a Tukey post hoc test. For comparisons between two groups, we used Student’s *t*-test. A *p*-value of 0.05 or less was considered significant.

## 3. Results

### 3.1. AGS and HIV Exposure Lowers Cathepsin Activities and Lysosome Numbers

To understand the mechanism(s) by which HIV and acetaldehyde induce accumulation of HIV proteins in hepatocytes, we measured cathepsin B and L activities as indices of lysosomal enzyme function. The combination of AGS and HIV treatment (AGS + HIV) decreased (*p* < 0.05) cathepsin B and L activities (Figure 1A,B). To determine whether lower cathepsin B and L activities by AGS + HIV exposure reflected lower lysosome numbers, we measured LAMP1 immunofluorescence intensity. Images were normalized to the number of cells in each field. LAMP1 intensity was 42% lower in AGS-treated uninfected cells, and this was further suppressed in HIV-infected cells (Figure 1C,D). Similarly, AGS decreased LAMP1 protein expression of HIV-infected cells as demonstrated on Western blots (Figure 1E,F). AGS exposure suppressed lysosome numbers in both HIV-infected and -uninfected RLW cells, and AGS + HIV suppressed lysosome functions in RLW cells.

### 3.2. Both AGS and HIV Enhance Lysosome Leakage

We investigated whether AGS-induced LAMP1 suppression affected lysosome membrane permeabilization and lysosomal leakage (LL) in HIV-infected and -uninfected cells. To this end, we quantified the levels of damage-regulatory autophagy modulator (DRAM)-1, which induces lysosome leakage, thereby triggering p53-dependent apoptosis [24]. Since we found that AGS treatment increased p53 mRNA levels in both infected and uninfected RLW cells (Figure 2A), we postulated that DRAM1 expressing cells would be cleared by apoptosis. In fact, it was observed that AGS and AGS + HIV exposed cells expressed a very low amount of DRAM1 (Figure 2B,C) because DRAM1-expressing cells were unaccounted for due to apoptosis. To prevent/slow the clearance of DRAM-positive cells, we measured DRAM expression after blocking apoptosis by a pan-caspase inhibitor (PCI). In the presence of PCI, we observed 64% higher levels of intracellular DRAM in AGS-exposed HIV-infected cells (Figure 2D,E). To further confirm these findings, we measured the co-localization of galectin 3 and LAMP1 puncta and found that AGS exposure exhibited greater co-localization of galectin 3 and LAMP1 puncta in HIV-infected cells than in uninfected cells (Figure 2F,G), indicating that both HIV and acetaldehyde exposure contributes to lysosome leakage.

### 3.3. AGS and HIV Enhance Intracellular Oxidant Stress

We investigated the contribution of oxidant stress to AGS-induced decline of cathepsin activities in control and HIV-exposed RLW cells as well as downstream apoptosis. Combined AGS and HIV exposure suppressed cathepsin B and L activities and caused an increase in apoptosis, detected by ELISA as elevated levels of M30 (cleaved cytokeratin 18) in cell supernatants. M30 detection was significantly lower (*p* < 0.05) after treatment of cells with N-acetyl cysteine (NAC), which restored cathepsin B and L activities (Figure 3A–C).

### 3.4. AGS and HIV Enhance Cathepsin Leakage and Intrinsic Apoptosis

We measured the leakage of cathepsin B from lysosomes and its translocation to the outer mitochondrial membrane, which likely damages the mitochondrial membrane. In AGS + HIV-treated cells, cathepsin B was co-localized with mitochondria, as indicated by immunofluorescent co-localization of cathepsin B and Tom20. Tom20 is a protein on the mitochondrial outer membrane that facilitates the import of mitochondrial precursor proteins into the mitochondrion [25,26] (Figure 4A,B). Furthermore, caspase 3 cleavage in AGS-exposed HIV-infected liver cells (the condition with the highest cathepsin B–Tom20 co-localization and apoptotic effects) was suppressed nearly three-fold in the presence of 20 µM caspase 9 inhibitor, while the caspase 8 inhibitor had a minimal effect on the observed apoptosis (Figure 4C,D).

### 3.5. AGS and HIV Exposure Suppress TFEB Protein Expressions in Cytosolic and Nuclear Fractions of RLW Cells

Lysosome damage reportedly activates lysosome biogenesis as a compensatory mechanism to replace damaged lysosomes [27] via enhanced TFEB transcription [15]. After we separated nuclear and cytosolic fractions from the control and treated RLW cells, we observed greater amounts of immunoreactive TFEB in the cytosolic fractions of HIV- and AGS-treated cells. Both treatments significantly decreased the nuclear to cytosolic TFEB ratios compared with untreated cells (Figure 5A,B).

The purities of our nuclear and cytosolic fractions were confirmed by Western blotting by exposing both fractions to anti-α-tubulin (a cytosolic marker) and to the anti-TATA binding protein (a nuclear marker) (Figure 5C).

Suppression by AGS of nuclear TFEB expression in HIV-infected and -uninfected cells prompted us to analyze the role of ethanol metabolism on the factors that regulate TFEB cytosolic-nuclear shuttling. As reported, mTORC1-mediated TFEB phosphorylation is an upstream event, which causes TFEB retention in the cytosol [15]. Surprisingly, we observed the downregulation of TFEB phosphorylation (pTFEB) at the mTOR-regulated serine sites (S211) in both AGS-exposed uninfected and HIV-infected cells, which decreased the pTFEB/TFEB ratio (Figure 5D,E). To test whether microtubules regulate TFEB trafficking to the nucleus, RLW cells were exposed for 24 h to 10 μΜ nocodazole, a microtubule-disrupting drug. We found that TFEB nuclear translocation was attenuated by nocodazole, indicating that translocation is microtubule-dependent (Figure 5F,G). We also found that exposure of HIV-infected and -uninfected RLW cells to AGS enhanced α-tubulin acetylation (Figure 5H,I).

### 3.6. AGS and HIV Exposures Suppress Proteasome Activity and Enhance ZKSCAN3 Expression

Because TFEB is degraded by the proteasome, we examined whether HIV or AGS influences 20S proteasome peptidase activities. HIV infection suppressed chymotrypsin and trypsin-like activities (Figure 6A,B). However, this effect was exacerbated in AGS-exposed HIV-infected and -uninfected RLW cells.

We explored whether acetaldehyde influenced TFEB induced activation of lysosomal genes. To this end, we measured the levels of ZKSCAN3 in RLW nuclear fractions as ZKSCANS represses TFEB-activated lysosome biogenesis. We observed a 40% rise in ZKSCAN3 expression after AGS treatment in both HIV-exposed and -unexposed RLW cells (Figure 6C,D). To corroborate these findings, we quantified the mRNAs encoding LAMP1 and cathepsin D, two genes activated by TFEB. Both LAMP1 and cathepsin D mRNAs were lower after cells were treated with AGS. This suppression effect was even greater in AGS + HIV-infected cells (Figure 6E).

## 4. Discussion

As revealed from our previous studies, pre-exposure of hepatocytes to the ethanol metabolite acetaldehyde sensitizes cells to HIV-induced apoptotic cell death. This was partly due to acetaldehyde-induced intracellular accumulation of HIV proteins, which occurs due to suppression of HIV protein degradation by both the proteasome and lysosomes [7]. In fact, inhibition of protein-degrading enzyme activities with specific inhibitors, MG132 and carfilzomib for proteasomes or bafilomycin and chloroquine for lysosomes, prolongs the persistence of HIV proteins in hepatocytes [7]. As reported by others, HIV proteins are degraded by proteasomes and lysosomes [28,29,30,31]. We showed that the accumulation of HIV proteins in hepatocytes was greater after treatment with lysosome inhibitors than with proteasome inhibitors. These findings underscore the importance of ethanol- and HIV-induced lysosomal dysfunction/damage for HIV protein retention, which can cause oxidative stress. In kinetic studies, we indeed observed that the accumulation of HIV proteins after exposure to AGS induces oxidative stress accompanied by ROS release, which leads to apoptotic hepatocyte death [7]. The latter event results in the formation of apoptotic bodies, which are engulfed by non-parenchymal (Kupffer) cells as we observed in some canonical hepatotropic infections, such as HCV-infection, and were able to further confirm in HIV infection [21,32].

Here, we addressed the contribution of lysosomal impairment to HIV alcohol-metabolite-induced hepatotoxicity. Our results lead us to suggest that acetaldehyde induces lysosomal damage in HIV-infected hepatocytes and that leakage of lysosomal luminal content (cathepsins) to other cell compartments may damage adjacent organelles, which hastens hepatic cell death. For example, leakage of cathepsins to mitochondria can disrupt the mitochondrial membrane and may induce intrinsic apoptotic events [27].

We focused on the underlying mechanism(s) by which HIV, ethanol metabolite(s), and their combination impaired lysosome activity. To mimic ethanol metabolism in hepatocytes, we exposed CYP2E1-expressing RLW cells to AGS. Lysosome leakage (LL) is one mechanism that leads to a decline in lysosome hydrolytic potency. The pathogenic impact of LL in the liver has already been discussed for non-alcoholic [33,34] and alcoholic steatohepatitis [35]. It was also implicated in the liver injury of ethanol-fed rodents [35,36]. However, this is the first time it has been investigated in non-immune cells (hepatocytes) after HIV infection and treatment with ethanol metabolites. In our hands, the treatment of both HIV-infected and -uninfected RLW cells with AGS lowered the expression of LAMP1, a lysosomal membrane protein important for maintaining lysosome structural integrity. These findings indicate that acetaldehyde exposure causes lysosomal membrane instability. The decrease in lysosomal numbers and functions were, at least in part, triggered by ethanol metabolite-induced oxidative stress since lysosome instability was attenuated by treatment with the antioxidant NAC. In fact, lysosomes do not express anti-oxidative enzymes, and they are sensitive to the effects of ROS, which easily destabilize lysosomal membranes [27]. Indeed, ROS have been shown to trigger LL in alcoholic steatohepatitis [34].

In the current study, co-localization of LAMP1 and galectin 3 reported by others as an indication of increased lysosomal permeability [37] was highest in AGS-treated HIV-infected RLW cells. While the role of HIV-induced lysosomal damage has not been studied in hepatocytes before, in CD4^+^ lymphocytes, HIV potentiates LL by activating lysosome-co-localized protein, DRAM1, responsible for lysosome membrane permeabilization (LMP) [24]. Furthermore, DRAM1 regulates apoptosis by increasing the lysosomal localization of pro-apoptotic protein BAX [38] and has p53 as its downstream target [39]. Therefore, by measuring p53 gene activation in AGS + HIV-treated RLW cells, we identified a significant increase in p53 mRNA levels. Because p53-mediated apoptosis is caspase-dependent [40], to preserve DRAM1-expressing RLW cells, we tested the effects of HIV and AGS treatments on DRAM1 expression in the presence of pan-caspase inhibitor (PCI) and found increased levels of DRAM1 in these double-treated cells compared with untreated (control) cells and cells exposed to either HIV or AGS. These changes were not observed in the absence of PCI because DRAM1-expressing cells undergo robust caspase-dependent apoptosis. Thus, it is likely that by activating p53-mediated genes in hepatocytes, DRAM1 contributes to apoptosis induction, demonstrating the link between LL and apoptotic cell death.

The link between LL and HIV–AGS-induced apoptosis in hepatocytes is still unclear. When RLW cells were exposed to both HIV and AGS, leading to the highest levels of apoptotic cell death based on the results of M30 ELISA, we observed the localization of cathepsin B to mitochondria (specifically, to Tom20 [41]), indicating cathepsin B translocation to mitochondria may be pro-apoptotic. This translocation can trigger caspase 3 cleavage and subsequent apoptotic cell death via upstream activation of caspase 9 by cytochrome C’s release from permeable mitochondria. In our hands, inhibition of caspase 9 by the specific blocker significantly suppressed HIV–AGS-induced caspase 3 cleavage, and this effect was more profound than the contribution of caspase 8 inhibition to caspase 3 cleavage, suggesting that activation of the intrinsic apoptotic pathway plays a major role in HIV + AGS-induced hepatocyte death. This prompts us to suggest that further investigations are warranted to study the interdependence of lysosomal and mitochondrial leakages in the pathogenesis of HIV and ethanol-metabolism-induced hepatotoxicity. However, since this is a highly complex and important area, we did not plan to intensively investigate lysosome–mitochondrial interactions in the frame of this study, but we intend to detail this crosstalk in future experiments. While in other studies, the role of LL and mitochondrial dysfunction for apoptosis induction has already been demonstrated [42,43,44], it has never been implemented into the lysosome–mitochondrion axis driving HIV-ethanol-metabolism-induced hepatotoxicity.

There are multiple compensatory mechanisms that remove damaged lysosomes by autophagy/lysophagy [45,46,47,48,49,50,51]. Major repair mechanisms are usually related to lysosome biogenesis via TFEB [45]. Here, we found that TFEB translocation from cytosol to the nucleus is necessary to activate lysosome-biogenesis-regulating genes and that this was impaired in AGS-exposed, HIV-infected hepatic cells. Indeed, the attenuation of TFEB translocation by ethanol metabolism in hepatocytes was reported before [13,52,53,54], but not in the context of HIV liver studies.

To address the mechanisms by which AGS (mainly acetaldehyde) inhibits TFEB nuclear translocation in hepatocytes in the settings of HIV, we tested the factors which regulate TFEB stabilization in the cytosol.

A variety of proteins and kinases control the translocation of TFEB to the nucleus to activate lysosomal biogenesis [15,16,55,56]. In this regard, we explored the role of phosphorylation as a post-translational modification that determines TFEB cytosolic accumulation in acetaldehyde-treated cells. Thus, phospho-TFEB dimerizes with TFEB to promote the retention of TFEB in the cytosol [57], which we observed in AGS-exposed, HIV-infected hepatocytes. mTOR phosphorylates TFEB at multiple serine residues (S122, S142, and S211). However, despite TFEB retention in cytosol, we even found the downregulation of acetaldehyde-induced TFEB phosphorylation at S211 [15]. The partial explanation for this event might be related to the overexpression of galectins in acetaldehyde-exposed liver cells due to their enhanced damage, leading to suppression of mTOR [58]. However, while we have no evidence that acetaldehyde increases TFEB phosphorylation, some other downstream acetaldehyde-affected events (e.g., aggregation) might account for TFEB cytosolic retention.

Given that TFEB is a substrate for the proteasome, we measured proteasome activities in HIV-infected and -uninfected RLW cells exposed or not to AGS and found a concomitant downregulation of chymotrypsin and trypsin-like proteasome activities. While there is no evidence of synergistic interactions between HIV and AGS, this is a novel finding because, to our knowledge, there are no studies that demonstrate the effects of acetaldehyde on TFEB in relation to proteasome activity in HIV-infected hepatocytes. The most likely scenario in ethanol-treated hepatic cells is the degradation of non-ubiquitylated TFEB by 20S proteasome because high oxidative stress causes the dissociation of 19S from 20S proteasome, thereby limiting its ability to recognize non-ubiquitinated substrates by the 26S proteasome [9]. In HIV-noninfected liver cells, we and others [7,59,60,61,62] have previously reported that ethanol or acetaldehyde impair chymotrypsin-like and trypsin-like activities of 20S proteasome catalytic core. We cannot exclude that proteasome dysfunction under HIV–AGS cell exposure attenuates the degradation of total TFEB in the cytosolic compartment. This undegraded TFEB should translocate to the nucleus for gene activation if its trafficking is not blocked.

The trafficking of TFEB by microtubules has not been reported before. However, in hepatocytes, the translocation of some other transcription factors such as STAT3 and STAT5B from cytosol to the nucleus is suppressed by ethanol metabolism, which impairs their trafficking due to microtubule acetylation [63,64]. Here, we observed that nocodazole, an inhibitor of microtubule trafficking, suppressed TFEB translocation to the nucleus in RLW cells. These findings indicate that microtubules are necessary for TFEB translocation from cytosol to nucleus. While based on our findings, HIV by itself does not induce acetylation of microtubules, in infected and non-infected RLW cells, AGS exposure increases TFEB acetylation, which negatively affects TFEB trafficking to the nucleus, resulting in its cytosolic retention.

Overall, the alcohol-impaired lysosomal biogenesis in HIV-infected hepatic cells is multifactorial, including the reduced translocation of TFEB to the nucleus. The latter may be attributed to decreased degradation of non-ubiquitylated TFEB by 20S proteasome in cytosol combined with acetaldehyde-impaired TFEB trafficking to the nucleus. Additionally, exposure of RLW cells to AGS increases the expression of master repressor ZKSCAN3 that blocks TFEB target gene activation, thereby suppressing lysosomal genes. To our knowledge, this transcriptional repressor of lysosome genes has never been explored in the context of the combined treatment of hepatocytes with acetaldehyde and HIV.

It became clear from our results and discussion that some downstream events we addressed in this study were a result of synergistic/additive effects of HIV with acetaldehyde. However, there are a lot of effects that were related to the properties of acetaldehyde without synergizing with HIV; while not synergistic/additive, they are still very important in mimicking the role of ethanol metabolism in the pathogenesis of HIV-induced liver injury in alcohol-abusing PLWH. To sort out the events which are “specific” for HIV-infected hepatocytes only and/or are more general in ethanol-metabolite-exposed liver cells was one of the purposes of this study, and the results are summarized in Table 1. Importantly, the changes in cathepsin activities, LL-related parameters (DRAM1 expression and LAMP1–Galectin 3 co-localization) as well as caspase 3 cleavage and TFEB translocation to nucleus was higher in AGS-exposed HIV-infected than in non-infected liver cells and are the results of synergistic or additive effects from both exposures—HIV and ethanol metabolites.

The proposed mechanism by which HIV and ethanol metabolism induce lysosome insufficiency contributing to hepatocyte cell death is summarized in Figure 7.

## 5. Conclusions

Findings from this study revealed acetaldehyde- and HIV-induced lysosomal leakage and TFEB dysregulation as the mechanisms of oxidative-stress-induced lysosome dysfunction in liver cells, which leads to hepatotoxic events. Therapeutic prevention of HIV–alcohol-triggered hepatotoxicity may be based on the identification and detailed characterization of these possible targets.

## Figures and Tables

**Figure 1 biomolecules-11-01497-f001:**
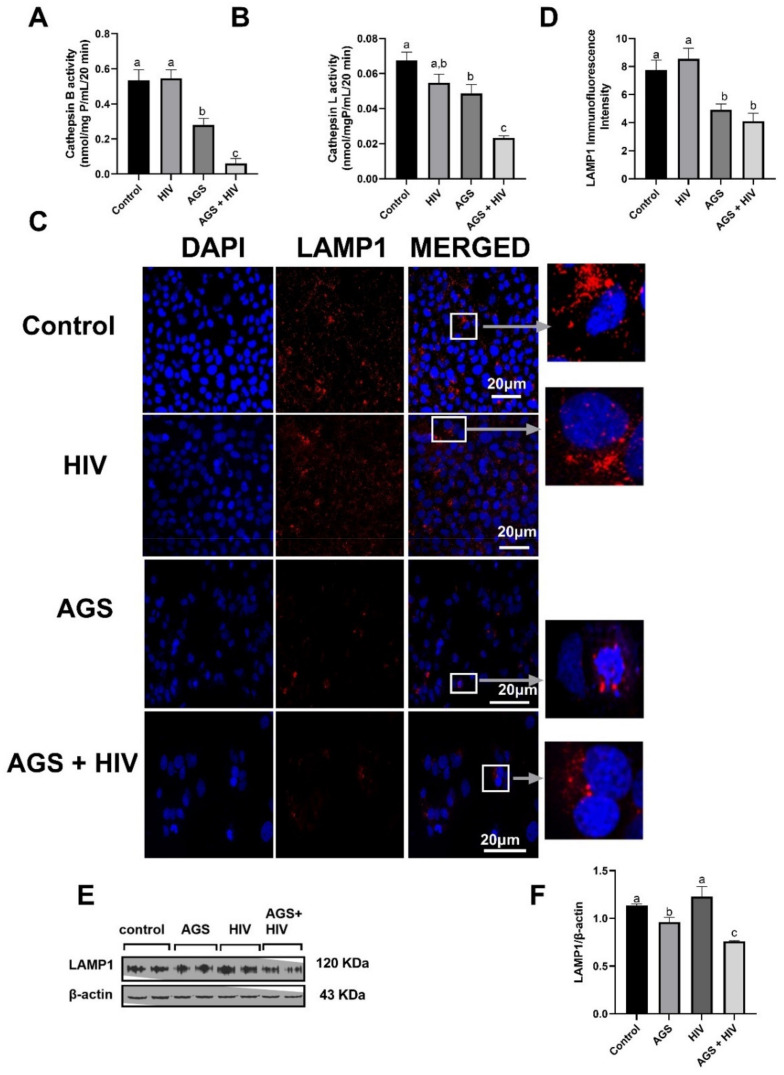
Acetaldehyde Generating System (AGS) exposure lowered cathepsin activities and lysosome numbers in HIV-infected and -uninfected RLW cells (RLW cells were treated as described in Materials and Methods). Specific activities of (**A**) Cathepsin B, and (**B**) Cathepsin L. (**C**) Immunofluorescent staining of LAMP1. Staining was visualized using a 63× lens in LSM 710 confocal microscope. Pictures are representative data from three independent experiments. (**D**) Quantification of LAMP1-stained lysosomes. LAMP1 immunofluorescence intensity was measured using ImageJ. (**E**,**F**) LAMP1 was measured by immunoblot analysis and quantification of immunoreactive protein bands. Data are from 3 independent experiments presented as mean ± SEM. Bars marked with the same letter are not significantly different; bars with different letters are significantly different from each other (*p* ≤ 0.05).

**Figure 2 biomolecules-11-01497-f002:**
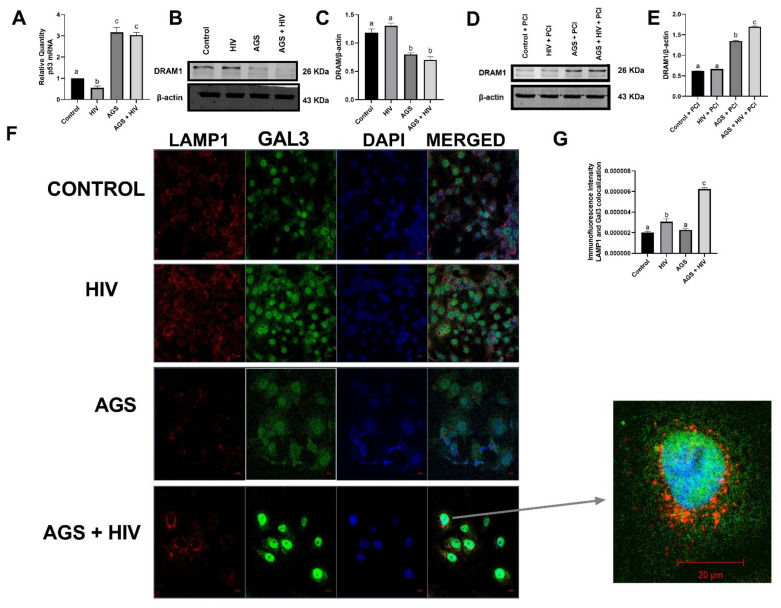
AGS and HIV enhance lysosomal leakage in RLW cells treated as indicated in Materials and Methods. (**A**) p53 mRNA levels were measured by RT-PCR with GAPDH mRNA as the internal standard. (**B**,**C**) DRAM1 protein expression in the absence of pan-caspase inhibitor (PCI) was detected by immunoblotting. Quantification of protein bands from 3 independent experiments is presented. Equal (20 µg) amounts of protein were loaded in each lane (as stated in Materials and Methods). β-actin was used as the internal standard. (**D**,**E**) DRAM1 protein expression in the presence of PCI was detected by immunoblotting. Quantification of protein bands from 3 independent experiments is presented. (**F**) Co-localization of galectin 3 and LAMP1 was measured by immunofluorescence. Proteins were visualized using a 40× lens LSM 710 confocal microscope. (**G**) Co-localization of galectin 3 and LAMP1 was quantified using NIH Image J. Data are from 3 independent experiments presented as mean ± SEM. Bars marked with the same letter are not significantly different from each other; bars with different letters are significantly different from each other (*p* ≤ 0.05).

**Figure 3 biomolecules-11-01497-f003:**
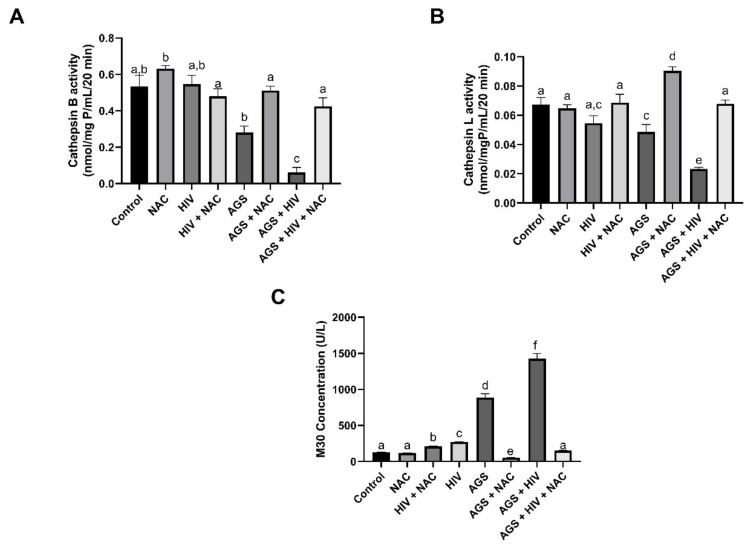
N-acetyl cysteine treatment prevents/reverses AGS and HIV-induced lysosomal leakage (RLW cells were treated as indicated in Materials and Methods). (**A**,**B**) Cathepsin B and Cathepsin L activities were measured in cell lysates (in the presence or absence of NAC) by fluorometric assay, using fluorimetric substrates. (**C**) Apoptosis was measured by M30 levels in cell culture supernatants. Data are from 3 independent experiments. Bars marked with the same letter are not significantly different from each other; bars with different letters are significantly different from each other (*p* ≤ 0.05).

**Figure 4 biomolecules-11-01497-f004:**
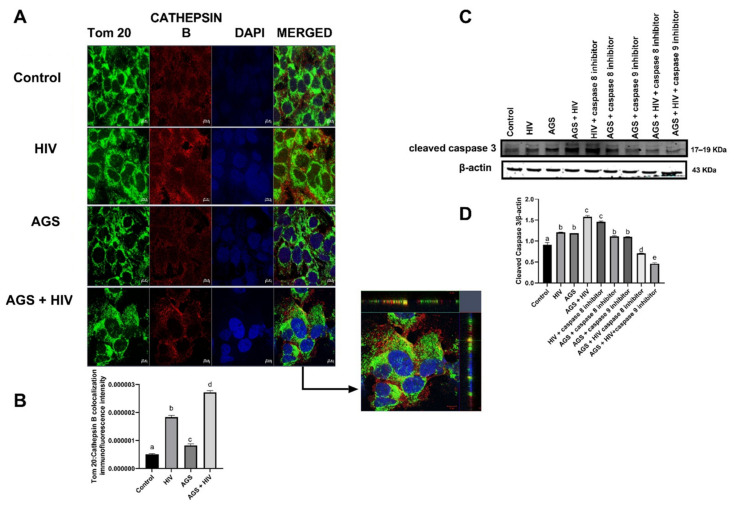
AGS and HIV enhance cathepsin B-triggered caspase 3-dependent apoptosis (RLW cells were treated as described in Materials and Methods) (**A**,**B**) Immunofluorescence detection of co-localization of Cathepsin B and Tom20 and quantification of staining. Staining was visualized using a 40× lens in LSM 710 confocal microscope, and staining intensity was quantified using NIH ImageJ. (**C**,**D**) Cleaved caspase-3 was measured by immunoblot analysis and quantification of immunoreactive bands from 3 independent experiments. Data are from 3 independent experiments presented as mean values ± SEM. Bars marked with the same letter are not significantly different from each other; bars with different letters are significantly different from each other (*p* ≤ 0.05).

**Figure 5 biomolecules-11-01497-f005:**
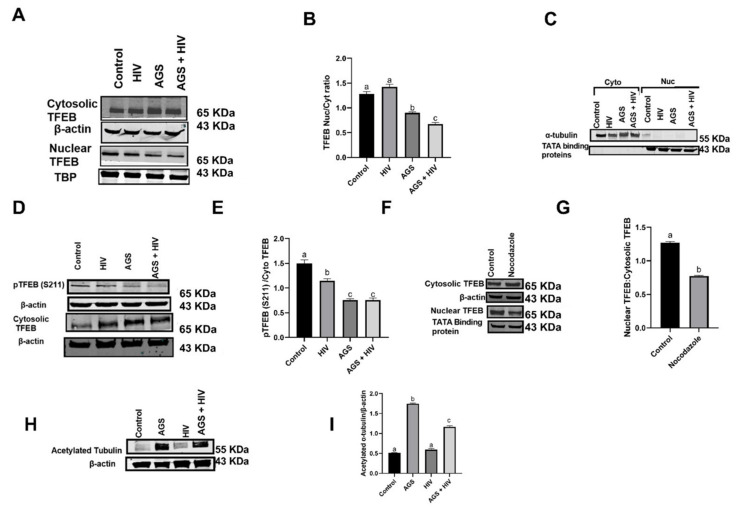
Suppression by AGS and HIV of transcription factor EB (TFEB) protein expression in cytosolic and nuclear fractions of RLW cells. (**A**) nuclear and cytosolic TFEB protein expressions were measured by immunoblot analysis. Equal (20 µg) amounts of protein were loaded in each lane. β-actin and TATA-binding proteins (TBP) were used as internal controls. (**B**) Quantification of nuclear and cytosolic TFEB ratios from 3 independent experiments. (**C**) Immunoblot analysis to confirm the purity of cytosolic and nuclear fractions. Equal amounts of cytosolic and nuclear proteins were loaded onto gels and blotted onto nitrocellulose. α-tubulin was checked in nuclear fractions and TBP in cytosolic fractions. (**D**) Immunoblotting analysis of phospho TFEB serine 211 protein expressions. Equal (20 µg) amounts of protein were loaded in each lane. β-actin and total TFEB were used as an internal control. (**E**) Quantification of serine 211 phosphorylation/total TFEB ratios. Data are from 3 independent experiments, presented as the mean ± SEM. Bars marked with the same letter are not significantly different from each other; bars with different letters are significantly different (*p* ≤ 0.05). (**F**) Nuclear and cytosolic TFEB was measured by immunoblotting analysis. Equal (20 µg) amounts of protein were loaded in each lane. β-actin and TBP were used as an internal control (**G**) Quantification of nuclear to cytosolic TFEB ratios from 3 independent experiments. (**H**,**I**) Acetylated tubulin was measured by immunoblot analysis and quantification of protein bands from 3 independent experiments. Data are from 3 independent experiments presented as mean ± SEM. Bars marked with the same letter are not significantly different from each other; bars with different letters are significantly different (*p* ≤ 0.05).

**Figure 6 biomolecules-11-01497-f006:**
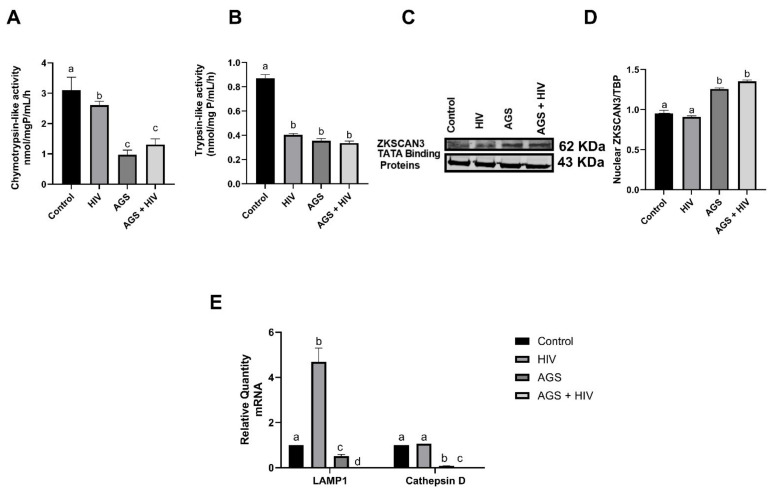
AGS and HIV enhance ZKSCAN3 but suppress TFEB target genes. (**A**,**B**) Cht-L, and T-L proteasome peptidase activities were detected in cell lysates by in vitro assay using fluorimetric substrates (**C**) Immunoblot analysis of ZKSCAN3 protein expression in RLW nuclear fraction. Equal (20 µg) amounts of protein were loaded in each lane. β-actin was used as an internal control. (**D**) Quantification of immunoreactive bands of ZKSCAN3 from 3 independent experiments. (**E**) LAMP1 and cathepsin D mRNA expression detected by RT-PCR analysis. GAPDH mRNA was used as an internal control. Data are from 3 independent experiments, presented as the mean ± SEM. Bars marked with the same letter are not significantly different from each other; bars with different letters are significantly different (*p* ≤ 0.05).

**Figure 7 biomolecules-11-01497-f007:**
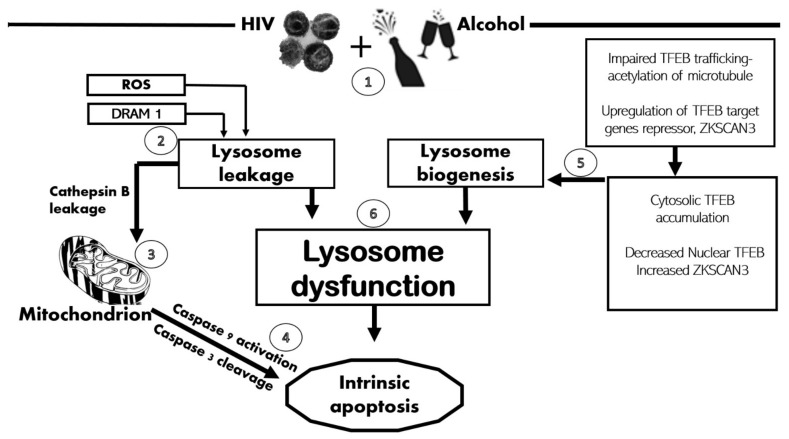
Role of lysosomal rupture/dysfunction in HIV- and ethanol-metabolism-induced apoptosis in hepatocytes: (1) Combined treatment with HIV and AGS triggers ROS and acetaldehyde release, which mediates lysosome leakage; (2) HIV/ethanol metabolism triggers the release of DRAM1, which also induces lysosome leakage; (3) Cathepsin B leaked out from damaged lysosome and diffused into the mitochondrion to initiate the intrinsic apoptotic pathway; (4) Caspases 9 and 3 become cleaved, leading to hepatocyte apoptosis; (5) Alcohol metabolites inhibit lysosome biogenesis factor TFEB, hence impairing the compensation of damaged lysosomes; and (6) Both lysosome damage and impaired lysosome biogenesis lead to HIV–ethanol-metabolism-induced lysosome dysfunction, which triggers apoptosis.

**Table 1 biomolecules-11-01497-t001:** Endpoints of the current study regulated by AGS only or by the combination of AGS/ethanol with HIV.

Parameters	Effect Due To:	Figures
Cathepsin B activity	AGS + HIV	Figure 1A
Cathepsin L activity	AGS + HIV	Figure 1B
LAMP1 immunostaining	AGS	Figure 1C,D
LAMP1 Western blot	AGS + HIV	Figure 1E,F
p53 mRNA	AGS	Figure 2A
DRAM1	AGS + HIV	Figure 2D,E
LAMP1 + Galectin 3 immunostaining	AGS + HIV	Figure 2F,G
Cathepsin B restoration by NAC	AGS + HIV—higher magnitude	Figure 3A
Cathepsin L restoration by NAC	AGS + HIV—higher magnitude	Figure 3B
M30	AGS + HIV	Figure 3C
Tom20 + Cathepsin B immunostaining	AGS + HIV	Figure 4A,B
Cleaved caspase 3	AGS + HIV	Figure 4C,D
Nuclear TFEB/Cyto TFEB	AGS + HIV	Figure 5A,B
PTFEB (S211)	AGS	Figure 5D,E
Acetylated Tubulin	AGS	Figure 5H,I
Chymotrypsin-like activity	AGS	Figure 6A
Trypsin-like activity	AGS	Figure 6B
ZKSCAN3	AGS	Figure 6C,D
LAMP1 mRNA	AGS + HIV	Figure 6E
Cathepsin D	AGS + HIV	Figure 6E

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
