# Peer review of "Alcohol-Induced Lysosomal Damage and Suppression of Lysosome Biogenesis Contribute to Hepatotoxicity in HIV-Exposed Liver Cells"

_biomolecules, 2021, doi:10.3390/biom11101497_

Round 1

Reviewer 1 Report

The authors have satisfactorily addressed most of the Reviewer's comments. However, paper still needs some minor corrections. The authors should carefully proofread the manuscript to correct some grammar errors

Reviewer 2 Report

All good

This manuscript is a resubmission of an earlier submission. The following is a list of the peer review reports and author responses from that submission.

Round 1

Reviewer 1 Report

The paper by New-Aaron et al. investigates the effect of ethanol metabolites in vitro on HIV-infected Huh7.5 cells and in vivo using humanized mice, focusing specifically on the effects on the lysosome and secondarily, apoptosis.  This paper addresses a clinically relevant and important problem, the effect of ethanol on the livers of people living with chronic HIV infections that are kept in check through anti retroviral therapy.  However, the mechanisms of alcohol-induced liver damage in people living with HIV are largely unknown.  Their central hypothesis is that exposure of hepatocytes to HIV and ethanol metabolites induces oxidant stress to trigger lysosomal leakage and suppresses lysosomal biogenesis, thereby inhibiting the replacement of damaged lysosome in hepatocytes.  This leads to increased hepatotoxicity under the conditions of the combined HIV and alcohol/acetaldehyde exposure to liver cells.

Overall, the conducted experiments are logical, and the manuscript is well written, however, a major conceptual concern is an apparent lack of any additive or synergistic effect between AGS and HIV in their cell culture model, and little to no additive or synergistic effect between ethanol and HIV in their in vivo model, in contrast to the hypothesis of “increased hepatotoxicity under the conditions of the combined HIV and alcohol/acetaldehyde exposure to liver cells”.

For example, in the in vitro data presented in Figure 1A and 1B, the effect on suppression of Cathepsin B and L activity (a proxy for lysosome function) is due solely to AGS.  There was no interaction between AGS and HIV to further suppress Cathepsin activity and HIV infection alone had no effect (except for a small increase in Cathepsin B activity).  A similar trend was also found for LAMP1 expression (Figure 1E and 1F), where AGS alone has a (modest) effect, HIV alone had little effect, and the combination was no greater than AGS alone.  In Figure 2A, the effect on p53 expression is driven by AGS, with no contribution from HIV.  Even when apoptosis is inhibited (with PCI), there was only a small increase in DRAM1 expression in AGS-treated HIV-infected cells.  In Figure 3C, HIV infection and AGS treatment increase M30 secretion (as a marker of cell death) to a similar degree, however, there was only a minor increase when the two were combined, demonstrating little to no additive effect of the two conditions.   Likewise, in Figure 5 and Figure 6, there was little additive effect between AGS and HIV treatment.  In the in vivo model, the effect on each endpoint appears to be driven by ethanol.  They do, however, see a small increase in oxidative stress when HIV-infected mice are placed on an ethanol diet. 

Therefore, the authors need to either conclusively demonstrate that ethanol can be a “second hit” to HIV infection, that exacerbates lysosomal-mediated apoptosis (studied end points), or adjust their conclusions to reflect the data.

Major technical concerns:

Overall, the immunofluorescence data need to be more carefully analyzed; and the quality for some Figures should be improved. 

  • In Figure 1C, the LAMP1 immunofluorescence does not appear to match the quantitation data in Figure 1D. For example, HIV-infected cells give a greater signal than Control, AGS, or AGS+HIV.  However, in Figure 1D, HIV is no different than Control.  Further in Figure 1C, some of the LAMP1 signal appears to be entirely nuclear. 
  • In Figure 2D, the signal from LAMP1 is very weak and impossible to interpret. The signal from DAPI is also very weak.  Therefore, the merged image is difficult to interpret.  Further, Galectin 3 expression appears to be mainly nuclear, whereas typically Galectin 3 is mostly cytoplasmic, with some expression in the nucleus.  In the paper cited by the authors (ref 40, Aits, et al) Galectin 3 is almost completely cytoplasmic. Also, in Figure 2C, control treatments (without PCI), should be included.
  • In Figure 4A, the purpose if this immunofluorescence panel was to demonstrate translocation (leakage) of Cathepsin B to the outer membrane of the mitochondria. However, there appears to be minimal overlap, and in fact, much of the cathepsin B signal is in the spaces between the TOM20 signal. 
  • In Figure 4C, cleaved Caspase-3 should have been measured in AGS and HIV-infected cells. Only the combination is shown.
  • Figure 7. In vivo The authors are encouraged to more thoroughly characterize the in vivo model, to at least include plasma ALT as a marker of liver injury (as mentioned above) and TUNEL for apoptosis. This is essential to determine if their model is working in that there is in fact liver damage resulting from ethanol feeding and if there is any additive or synergistic effect by a pre-existing HIV infection, the central premise of this study. 
  • Having only 3 mice/group seems to be too few to draw any conclusions due to the typically large variability seen in the NIAAA model. Need to justify why only 3 mice/group were used.  Should present Western blot of all mice, instead of 1 from each group (Figure 7A). 

Minor concerns:

  • Figure 1D, the X-axis does not appear to be labeled properly. Is the graph in Figure 1F an average of only the Western blot shown in Figure 1E, or an average of several experiments?
  • The Discussion section is too long. The authors should focus primarily on interpretation of the data and not restating the results. 
  • Although the manuscript is for the most part well written, there are still numerous instances of poor grammar. For example, line 20, “…metabolites impair lysosome to enhance…” and line 202 “deacrease”, to point out a few.  The authors are encouraged to carefully edit this manuscript for grammar and typos.

Author Response

We would like to thank the reviewers for their constructive comments. We are pleased with the estimation of this manuscript as an important, clinically relevant, and well-written, with logical experiments. However, there were some concerns, which we would like to address:

  1. major conceptual concernis an apparent lack of any additive or synergistic effect between AGS and HIV in their cell culture model, and little to no additive or synergistic effect between ethanol and HIV in their in vivo model, in contrast to the hypothesis of “increased hepatotoxicity under the conditions of the combined HIV and alcohol/acetaldehyde exposure to liver cells.”

Response:  We understand the reviewer’s concerns, and for better explanation of the complexity of HIV-1 and EtOH effects on liver pathology, we now added Table 1. Table 1 shows that there are no simple additive or synergistic effects. The obtained data strongly supports our conclusion about increased hepatotoxicity in HIV-infected hepatic cells exposed to ethanol metabolites. While some effects on lysosome come equally from exposure of both HIV-infected and non-infected RLW cells to AGS and thus, are attributed to the effects of acetaldehyde, other ones can be obtained only under combined cell exposure to both HIV and AGS. The latter in vitro detected changes are mainly related to the parameters characterizing lysosomal damage (DRAM1 expression and LAMP1-Gal3 co-localization), leakage of cathepsin B to mitochondria (Tom 20-Cathepsin B co-localization), cell apoptosis (M30 ELISA and cleaved caspase 3) as well as nuclear TFEB/cytosolic TFEB ratio and activation of lysosomal genes (LAMP1 mRNA), which are the central players for our hypothesis. The input of HIV exposure in the absence of AGS to lysosome function/downstream apoptosis is next-to-nothing. Still, in combination with AGS, HIV induces substantial changes, significantly exceeding the effects of AGS only (even if the magnitude of combined response is not very high, it is still substantially different from AGS). Based on our NAC data, it may be related to the overall level of oxidative stress, which partly comes from HIV and partly from AGS and only when combined, can reach the levels to cause lysosomal damage, leakage of lysosomal enzymes to mitochondria and activation of intrinsic apoptosis in liver cells. Now all data characterizing either the effects of AGS in HIV-infected cells or synergistic/additive effects of HIV+AGS in RLW cells are summarized in Table 1, Discussion. We run additional experiments with more replicates to re-check the effects of the combined treatments (cathepsin activities, M30). Updated results are presented in Figure 1 A, B and Figure 3 A, B, C.  Regardless of whether all other effects are synergistic/additive or not, AGS-induced changes even in the absence of synergism between acetaldehyde and HIV will be potentially observed in HIV-infected alcohol-abusing patients and contribute to the pathogenesis of liver injury in HIV-infected alcohol abusers.

  1. In Figure 1C, the LAMP1 immunofluorescence does not appear to match the quantitation data in Figure 1D. For example, HIV-infected cells give a greater signal than Control, AGS, or AGS+HIV.  However, in Figure 1D, HIV is no different than control.  Further, in Figure 1C, some of the LAMP1 signals appear to be entirely nuclear. 

Response: The density of cells at the end of treatments is not the same in all treated wells. We normalized the intensity of fluorescence to cell numbers in the field. That is why the quantitation data in Figure 2 D look a bit different. To confirm the localization of LAMP1, now we enlarged the images in Figure 1C, and you can see that LAMP1 localization is cytosolic.

  1. In Figure 2D, the signal from LAMP1 is very weak and impossible to interpret. The signal from DAPI is also very weak.  Therefore, the merged image is difficult to interpret.  Further, Galectin 3 expression appears to be mainly nuclear, whereas typically, Galectin 3 is cytoplasmic mainly, with some expression in the nucleus.  In the paper cited by the authors (ref 40, Aits, et al.) Galectin 3 is almost entirely cytoplasmic. Also, in Figure 2C, control treatments (without PCI) should be included.

Response: We equally adjusted brightness for the LAMP1 signal on all images in Figure 2D (now is Figure 2F). Regarding the localization of Galectin 3, the literature has already reported that in hepatoma cells, nuclear localization of Galectin 3 is possible, and our Huh 7.5 cells are of hepatoma origin.  However, for co-localization with LAMP1, we quantified cytosolic puncta, and we respectfully disagree with the reviewer that Gal3 localization is not cytosolic. As per the reviewer's request, we included data on DRAM1 expression without PCI (Fig. 2B, C).

  1. In Figure 4A, the purpose of this immunofluorescence panel was to demonstrate translocation (leakage) of Cathepsin B to the outer membrane of the mitochondria. However, there appears to be minimal overlap, and in fact, much of the cathepsin B signal is in the spaces between the TOM20 signal.

Response: The co-localization is visible as a yellow color. Both proteins are diffusely distributed in the cytoplasm as red (cathepsin) and green (TOM20) signals, and only co-localization provides the yellow signal. We quantified only co-localized Cathepsin with TOM 20 (Fig.4B). The expression of co-localized proteins cannot be high since such cells undergo cell death. We observed the same pattern with co-localization of Cathepsin B and VDAC (not shown) in HIV-infected RLW cells exposed to AGS.

  1. In Figure 4C, cleaved Caspase-3 should have been measured in AGS and HIV-infected cells. Only the combination is shown.

Response: In this paper, we showed the effects of caspase 9 and 8 inhibitors on caspase 3 cleavage induced by the combination of HIV and AGS because our previous studies demonstrated that the effects of HIV on caspase 3 cleavage are next-to-nothing, but there is a synergistic effect of HIV and AGS on caspase 3 cleavage (Ganesan et al., 2019, Biomolecules). Surprisingly, while HIV-infection of RLW cells in the absence of AGS induced very little caspase 3 cleavage, the treatment of HIV-infected cells with caspase 9 and 8 inhibitors provided unexpected effect inducing caspase 3 cleavage instead of blocking it. The doses of these inhibitors were not cytotoxic.  This phenomenon was very reproducible, but we could not interpret it. To avoid misleading the readers, we showed the effects of caspase 9 and 8 inhibitors on caspase 3 cleavage only in RLW cells exposed to the combination of HIV and AGS, resulting in the highest level of apoptotic cell death.

  1. Figure 7. In vivo, the authors are encouraged to more thoroughly characterize the in vivo model, to at least include plasma ALT as a marker of liver injury (as mentioned above) and TUNEL for apoptosis. This is essential to determine if their model is working in that there is liver damage resulting from ethanol feeding and any additive or synergistic effect by a pre-existing HIV infection, the central premise of this study.

Response: All this information was included in our previous paper (Ganesan et al., 2019, Biomolecules, PMID: 31835520). Here, we just used tissue of humanized mice from the already published ethanol feeding experiment to confirm the effects of HIV and ethanol on lysosomal disfunction.

  1. Having only 3 mice/group seems too few to draw any conclusions due to the typically large variability seen in the NIAAA model. Need to justify why only 3 mice/group were used.  Should present Western blot of all mice, instead of 1 from each group (Figure 7A). 

Response: Humanized mouse model is costly, and sometimes it is challenging to generate a large number of mice with the same level of human hepatocytes engraftment for four group experiments. However, as you can see from individual Western blot data, the results look very tight.   Now we show Western blot data for all tested mice (Figure 7A).

Minor concerns:

  1. Figure 1D, the X-axis does not appear to be labeled properly. Is the graph in Figure 1F an average of only the Western blot shown in Figure 1E, or an average of several experiments?

Response: We changed the labeling of the X-axis in Fig. 1 D. Western blot quantification is from this representative experiment, but we have several experiments showing the same pattern.

  1. The Discussion section is too long. The authors should focus primarily on the interpretation of the data and not restating the results.

Response: We have shortened the Discussion. Now it mainly focuses on interpreting the results; however, we also wanted to explain why these experiments have been done and how the obtained data can be incorporated in HIV -induced liver disease pathogenesis triggered by ethanol metabolism. 

  1. Although the manuscript is, for the most part, well written, there are still numerous instances of poor grammar. For example, line 20, “…metabolites impair lysosome to enhance…” and line 202 “decrease”, to point out a few.  The authors are encouraged to carefully edit this manuscript for grammar and typos.

Response: We apologize for the errors. Now we checked all spellings, and hopefully, minimized the misspellings. We would like to thank the reviewer for the comment that in general, this manuscript is well-written.

Reviewer 2 Report

In principle, this is an interesting angle to look at alcohol-HIV co-pathogenesis.

The following must be improved and re-reveiwed:

1) The measurements of TFEB by expression are unclear. This must be completely reworked.

2) ZKSCAN3 is completely out of place here, should be removed.

3) The mouse studies with liver humanization need more control than human albumin. Histological data must be properly presented.

4) What was the immune status of these mice?

4) The introduction, discussion and referencing is very unbalanced and must include these key references on lysosomal damage:

PMID: 27753622 / PMID: 28743755 / PMID: 30314966 / PMID: 31813797 PMID: 31995728 / PMID: 24100292 / PMID: 29625033

5) The choice of HIV1 ADA - M-tropic needs to be explained.

Author Response

We thank the reviewer for estimating this study as an interesting angle to look at alcohol-HIV pathogenesis.

  1. The measurements of TFEB by expression are unclear. This must be completely reworked

Response: Based on the reviewer's suggestion, we reworked the TFEB expression.

  1. ZKSCAN3 is completely out of place here, should be removed

Response: We still want to keep ZKSCAN3 because we are talking about the effects of ethanol metabolism on lysosomal biogenesis in HIV-infected cells in this part of the manuscript. ZKSCAN3 is a repressor for gene activation. Thus, ethanol metabolism's negative “influence” on lysosomal gene induction may come both from impaired translocation of TFEF to the nucleus and activation of this repressor of TFEB-triggered signaling.

  1. The mouse studies with liver humanization need more control than human albumin. Histological data must be properly presented

Response: We included an additional reference that described the morphology of animal liver tissues in the previously published paper. Ganesan M, New-Aaron M, Dagur RS, Makarov E, Wang W, Kharbanda KK, Kidambi S, Poluektova LY, Osna NA. Alcohol Metabolism Potentiates HIV-Induced Hepatotoxicity: Contribution to End-Stage Liver Disease. Biomolecules. 2019 Dec 10;9(12):851. doi: 10.3390/biom9120851. PMID: 31835520; PMCID: PMC6995634. Readers will be able to find additional characteristics of these humanized mice: staining for human-specific cytokeratin, Caspase-3, and Ki-67 as markers for apoptotic death and proliferation. We cannot duplicate already published information.

  1. What was the immune status of these mice?

Response: Like all types of humanized mice, these mice are immunodeficient. That is why there were not infected but injected with HIV.

  1. The introduction, discussion and referencing is very unbalanced and must include these key references on lysosomal damage: PMID: 27753622 / PMID: 28743755 / PMID: 30314966 / PMID: 31813797 PMID: 31995728 / PMID: 24100292 / PMID: 29625033

Response: We thank the reviewer for this helpful suggestion. We adjusted Introduction and Discussion, and now the suggested references are cited.

  1. The choice of HIV1 ADA - M-tropic needs to be explained

Response: It is not known which subtype of HIV is hepatotropic. The infection may be brought to the liver by immune cells, and liver macrophages as non-parenchymal liver cells are an important barrier playing a leading role in the gut-liver axis. These macrophages can be infected with HIV ADA. We wanted to know how the macrophage-tropic virus affects hepatocytes, which are in close contact with both resident and circulating macrophages.

Round 2

Reviewer 1 Report

The study by New-Aaron et al. investigates the effect of ethanol metabolites in vitro on HIV-infected Huh7.5 cells and in vivo using humanized mice, focusing specifically on the effects on the lysosome and secondarily, apoptosis.  The revised manuscript has some improvements, but several issues still remain and require significant attention: 1) lack of proper controls in cell culture experiments to ensure scientific rigor and reproducibility; 2) some images and their quantification are still misleading; 3) significant concerns regarding animal study and overinterpretation of the data. In opinion, of this reviewer, the author should perform a major revision of their manuscript. Below, I provide 3 major comments that were not properly addressed by the authors.

Reviewer comment from the 1st revision:

In Figure 4C, cleaved Caspase-3 should have been measured in AGS and HIV-infected cells. Only the combination is shown.

The authors' response to this basic request for appropriate controls was to indicate that a previous study demonstrated that HIV infected cells had "next-to-nothing" cleavage of caspase 3.  They further went on to indicate that other controls (HIV + caspase 8 or caspase 9 inhibitors), actually induced caspase 3 cleavage, but they decided to leave these data out so as not to mislead readers. However, to ensure scientific rigor and reproducibility, all experiments should be controlled for all variables, regardless of any previous experiment that may have also included the same control.  Otherwise, correct interpretation of the data becomes impossible due to inherent variations between experiments.  Therefore, the authors must include the following controls: AGS alone, HIV alone, AGS + caspase inhibitors, and HIV + inhibitors.  Leaving out data because no explanation can be formulated is not appropriate.

Reviewer comment from the 1st revision:

In Figure 1C, the LAMP1 immunofluorescence does not appear to match the quantitation data in Figure 1D.  For example, HIV-infected cells give a greater signal than Control, AGS, or AGS+HIV.  However, in Figure 1D, HIV is no different than Control.  Further in Figure 1C, some of the LAMP1 signal appears to be entirely nuclear. 

The authors state that this can be explained by the density of the cells.  However, a visual assessment does not support this notion. If one even looks at individual cells, HIV-infection clearly increases LAMP1, whereas AGS, nor AGS+HIV has any effect and looks exactly like control cells.  Therefore, the proper images matching the quantitation data are strongly recommended. In addition, it seems that the author has recently demonstrated the similar experimental concept in Fig.4f (PMID: 33466299). The author can reference that paper to avoid any misinterpretation of the data.

Reviewer comment from the 1st revision:

Figure 7.  In vivo experiment.

No ALT data is reported for the in vivo model.  This is essential to determine if their model is working in that there is in fact liver damage resulting from ethanol feeding and if there is any additive or synergistic effect by a pre-existing HIV infection, the central premise of this study.  Having only 3 mice/group seems to be too few to draw any conclusions due to the typically large variability seen in the NIAAA model.  Need to justify why only 3 mice/group were used.  Typically, a minimum of 8 mice/group is necessary owing to the significant mouse-to-mouse variability.

The authors are encouraged to more thoroughly characterize the in vivo model, to at least include plasma ALT as a marker of liver injury (as mentioned above) and TUNEL for apoptosis, and blood alcohol levels.

Should present Western blot of all mice, instead of 1 from each group (Figure 7A). 

There are several issues with their response to this very legitimate and important concern regarding this animal experiment. First, it was not clear from the original draft of the manuscript that the mouse experiment was described in a previous publication (PMID: 31835520).  This should have been clearly pointed out and referenced. The brief description of the phenotype should be provided in the current manuscript to properly interpret the new data. For one, there was no increase in plasma ALT in ethanol-fed mice.  Secondly, there were no statistically significant effects by HIV+ethanol (and there were only 2 mice in that group, which is too few for any statistically analysis).  Therefore, any conclusions based upon these mice are not supported by data.  We also suggested performing TUNEL analysis.  They indicated that "All this information was included in our previous paper.", which it clearly was not.  In the previous paper to which they refer (PMID:31835520), an analysis of Casp-3 by immunohistochemistry was performed, but was not quantitated, so no conclusions can be drawn from that piece of data.  It seems, there was very little interaction between ethanol and HIV in promoting liver injury in vivo other than a small increase in MDA (of note, it is unclear if Fig., 7G in the current manuscript is the same data presented in different way in the PMID:31835520).  Overall, the data are overinterpreted and conclusions are not supported by the data.  In opinion of this reviewer, removing the animal study will significantly improve the current manuscript.

Author Response

Thank you for thorough review of our manuscript. We have redone several experiments, and we do hope that now all presented data would satisfy the reviewers.

  1. The authors must include the following controls: AGS alone, HIV alone, AGS + caspase inhibitors, and HIV + inhibitors. Leaving out data because no explanation can be formulated is not appropriate.

Response: We re-run these blots, and now all requested controls are presented on the blot. As we mentioned before, we cannot provide an explanation why caspase inhibitors even increased caspase 3 cleavage under HIV only treatment (which in the absence of these inhibitors did not provide robust apoptosis; this is not related to toxic effects of these concentrations of inhibitors, which was already tested by us). However, we observed very significant effect protective of caspase 9 inhibitor under AGS+ HIV conditions, where caspase 3 cleavage was the highest.

  1. In Figure 1C, the LAMP1 immunofluorescence does not appear to match the quantitation data in Figure 1D. For example, HIV-infected cells give a greater signal than Control, AGS, or AGS+HIV. However, in Figure 1D, HIV is no different than Control. Further in Figure 1C, some of the LAMP1 signal appears to be entirely nuclear.

Response: In the previous version of Fig.1 C, cell density was not the same under different treatments, and because of this, the quantification was not obvious (we normalized fluorescence to cell number in the field). Now we re-run these experiments, and hope that now the changes in LAMP1 expression in RLW cells become more visible. The enlarged images from Fig. 1C clearly demonstrate that the localization of LAMP1 is cytosolic. The quantification of immunofluorescent images is fully consistent with the results of western blot and confirms what we got in Fig. 4F (PMID: 33466299) in different experiments from our previously published paper, where in immunofluorescent image the results of treatment with HIV and AGS were mistakenly switched, but the result of HIV+AGS was treatment was the same.

  1. Figure In vivo experiment. No ALT data is reported for the in vivo model. This is essential to determine if their model is working in that there is in fact liver damage resulting from ethanol feeding and if there is any additive or synergistic effect by a pre-existing HIV infection, the central premise of this study. Having only 3 mice/group seems to be too few to draw any conclusions due to the typically large variability seen in the NIAAA model. Need to justify why only 3 mice/group were used. Typically, a minimum of 8 mice/group is necessary owing to the significant mouse-to-mouse variability. The authors are encouraged to more thoroughly characterize the in vivo model, to at least include plasma ALT as a marker of liver injury (as mentioned above) and TUNEL for apoptosis, and blood alcohol levels. Overall, the data are overinterpreted and conclusions are not supported by the data. In opinion of this reviewer, removing the animal study will significantly improve the current manuscript

Response: Following the recommendation of the reviewer, we removed animal data from the current manuscript. Unfortunately, liver tissue we had from our prior animal study was very limited and too old to perform TUNEL analysis, while we still were able to use it for western blot analysis of LAMP1 and cathepsin-proteasome activities assays.

We hope that this time, we alleviated the concerns of the reviewer regarding the data presentation for the current manuscript.